

# Fluorescent aerosol particles in the Finnish sub-Arctic during the Pallas Cloud Experiment 2022 campaign

Jürgen Gratzl[1], David Brus[2], Konstantinos Doulgeris[2], Alexander Böhmländer[3], Ottmar Möhler[3], Hinrich Grothe[1]

5  [1]Institute of Materials Chemistry, TU Wien, Vienna, 1060, Austria
[2]Finnish Meteorological Institute, Atmospheric Composition Research, Helsinki, FI-00560, Finland
[3]Institute of Meteorology and Climate Research, Atmospheric Aerosol Research (IMK-AAF),
Karlsruhe Institute of Technology (KIT), Karlsruhe, 76021, Germany

*Correspondence to*: Hinrich Grothe (hinrich.grothe@tuwien.ac.at)

10  **Abstract.** Fluorescent aerosol particles (FAPs) as a fraction of total aerosol particles (TAPs) were measured online with a Wideband Integrated Bioaerosol Sensor 5/NEO (WIBS, Droplet Measurement Technologies) from mid-September to mid-December during the Pallas Cloud Experiment 2022 (PaCE22) at the Sammaltunturi station, located in the sub-Arctic region of Finnish Lapland. The WIBS measures particle size distributions from 0.5 to 30 µm and fluorescence in three channels of single aerosol particles, as well as particle concentrations. Since most biological aerosol particles exhibit intrinsic fluorescence, 15  FAP concentration can be used as a proxy for primary biological aerosol particles (PBAPs) like bacteria, fungal spores and pollen. The concentrations and size distributions of different fluorescent particles, together with meteorological data and air mass trajectories allow valuable insights to the emission of PBAPs from northern boreal forests and their dynamic in the atmosphere. We found a clear seasonal trend for most FAP types and a strong, sudden decrease in concentration after the surrounding ground is covered in snow. Caution should be taken in interpreting the data, as interference may be introduced by 20  non-biological fluorescent particles like secondary organic aerosols or soot, as well as biological secondary organic aerosol. The data is available at the open data repository Zenodo under the doi 10.5281/zenodo.13885888 (Gratzl and Grothe, 2024).

## 1 Introduction

The atmosphere is a dynamic reservoir of diverse aerosol particles, influenced by both anthropogenic and biological sources. 25  Aerosol particles play a vital role in the radiative balance of earth's atmosphere by scattering and absorbing sunlight, either directly or by changing cloud albedo and cloud lifetime (Lohmann and Feichter, 2005). A special subset of aerosols contains primary biological aerosol particles (PBAPs) which include, but are not limited to airborne viruses, bacteria, fungal spores, pollen grains and plant debris (Després et al., 2012). Due to the interaction of PBAPs with human, animal and plant health (e.g. disease transmission, allergic reactions, crop diseases), especially fungal spores and pollen grains have been monitored 30  for decades by aerobiologists using traditional methods like the Hirst trap, first introduced in 1952 (Hirst, 1952). This method





relies on the capture of PBAPs on a slowly moving sticky microscope slide and subsequent analysis of pollen grains and fungal spores under an optical microscope. Other commonly used techniques involve examining PBAP concentrations with fluorescence microscopes after DNA staining of PBAPs on filters or by incubation on Agar plates (Després et al., 2012). However, these methods have limited time resolution, and require trained personnel, as well as a high expenditure of time to

identify PBAPs.

Only in recent years PBAPs have gained more attention from atmospheric scientists, partly because a new technology was commercially available since the late 1990s, i.e. Laser/Light induced Fluorescence (LiF) of single aerosol particles. The first commercially available instrument was the UV-APS from TSI, first described by Hairston et al. (1997) and Brosseau and Hairston (2000). Another commonly used instrument is the Wideband Integrating Bioaerosol Sensor (WIBS) from Droplet

Measurement Technologies (DMT), which is used in this study. These instruments detect the intrinsic fluorescence of single aerosol particles, enabling the measurement of fluorescent aerosol particles with high time resolution and in near-real time. Particles of biological origin often contain bio-fluorophores like tryptophan, nicotinamide adenine dinucleotides (NAD(P)H) and riboflavin (Pöhlker et al., 2013) which can be detected using suitable excitation and emission wavelengths. Therefore, fluorescent aerosol particles (FAPs) can be interpreted as a proxy for PBAPs (Artaxo et al., 2022). For example, several field

studies showed a good correlation between LiF-Instrument-obtained particle concentrations and fungal spores or fungal spore-tracers in the field (e.g. Healy et al., 2014, Fernández-Rodríguez et al., 2018, Gosselin et al., 2016, Yue et al., 2019, Yue et al., 2022, Markey et al., 2022).

Beyond an aerobiology and immunology perspective, bioaerosols are also studied in the context of cloud formation. Several

laboratory- and field studies (for an overview, see Duan et al., 2023, Kanji et al., 2017, Fröhlich-Nowoiky et al., 2016,  Després et al., 2012, Hoose and Möhler, 2012, Möhler et al., 2007) showed that biological particles are  effective known ice nucleating particles (INPs), triggering the glaciation of water droplets at very high sub-zero temperatures: For example, the bacteria *Pseudomonas syringe* proofed to nucleate ice at -2°C (Maki et al., 1974) making it the most effective known ice nucleator. Besides bacteria, several atmospheric relevant fungal spores (Morris et al., 2013, Haga et al., 2013, Haga et al., 2014, Fröhlich-

Nowoisky et al., 2015) and pollen grains (Pummer et al., 2012, Diehl et al., 2001) as well as their associated nanometer-sized macromolecules (Pummer et al., 2012, Augustin et al., 2013, Pummer et al., 2015, Wieland et al. 2024) can nucleate ice at high or moderate sub-zero temperatures. Other potentially atmospherically relevant bioparticles like plant debris (Puxbaum et al., 2003) or subpollen particles/pollen fragments (Taylor et al., 2004) can also be ice active (Schnell and Vali, 1976, Seifried et al., 2023, Burkart et al., 2021, Matthews et al., 2023). Thus, traditional methods of aerobiologists might not be suitable for

ice nucleation research, as they only detect intact and big fungal spores and pollen grains with low time resolution and therefore potentially overlook important contributors to biological INPs, especially in the size range below about 2 µm in diameter (Fernández-Rodríguez et al., 2018). To better understand biological atmospheric ice nucleation, Lif-Instruments are increasingly used in field studies addressing the question if and how biological particles actually play an important role in cloud glaciation and therefore precipitation (Fennelly et al., 2018). For instance, a strong link of biological particles measured



with Lif-Instruments and warm INPs has been found in a boreal forest in southern Finland (Schneider et al., 2021) and in the high arctic (Pereira Freitas et al., 2023).

However, certain non-biological aerosols show weak fluorescence as well and can therefore bias the measurement: Savage et al. (2017) characterized several biological and non-biological particles in the laboratory and showed that for instance, PAHs (Polyaromatic Hydrocarbons), and certain types of soot and brown carbon introduce false positives. In field studies, a

correlation between eBC (elemental black carbon) mass concentrations and certain fluorescent particle types measured with the WIBS (see Table 1) was observed (Gao et al., 2024, Beck et al., 2023, Yue et al., 2022). In Table 1, possible contributors of biological and nonbiological aerosol particles that have been identified in both laboratory experiments and in field studies, are listed.

Pallas, as a clean continental background station with minimal anthropogenic influence, is an ideal site to study ambient

fluorescent aerosol particles (Doulgeris et al., 2022). By choosing an appropriate fluorescent background threshold (Savage et al., 2017) only minor non-biological interferences are expected. An Introduction/overview of the Pallas Cloud Experiment 2022 (PaCE 2022) campaign is given in Brus et al. (2025).

## 2 Observation Site

The instrument was installed at the Sammaltunturi station, part of the Pallas Atmosphere-Ecosystem Supersite in Finnish

Lapland, hosted by the Finnish Meteorological Institute. The station is located at 67.9733° N and 24.1157° E and approximately 170 km above the Arctic Circle. According to the Köppen climate classification (Köppen, 1931) the location is on the northern edge of the sub-Arctic climate zones (Beck et al., 2018). It is built at an altitude of 565 meters above sea level, at the summit of Sammaltunturi hill. The tree line is approximately 100 m below the station. Above the tree line, the vegetation is dominated by low vascular plants, lichen and moss (Hatakka et al., 2003). The boreal forest around the station consists of pine, spruce

and birch (Komppula et al., 2005). The station is located within the Pallas-Yllästunturi-National Park. Local or regional sources of air pollution are scarce. Therefore, Sammaltunturi is an important station to measure background air in northern Europe (Lohila et al., 2015) and is an ideal place to observe biological aerosol particles using single particle fluorescence measurements, as the surrounding boreal forest is an important source of PBAPs (Schumacher et al., 2013) and false positives (non-biological fluorescent particles) are minimized due to the clean air conditions.

## 3 Instrumentation

Near-real-time measurements of fluorescent and total aerosol particles were conducted with the Wideband Integrated Bioaerosol Sensor WIBS 5/NEO (DMT), referred to hereafter as WIBS. The WIBS operated with an inlet flow of 0.3 l min$^{-1}$ and an internally produced sheet flow of 2.1 l min$^{-1}$. A 635 nm Laser is used to measure the size of single particles via Mie scattering induced intensity reduction as well as a trigger signal for two xenon lamps firing at 280 and 370 nm used as a

fluorescence excitation source. The emitted fluorescence is collected with two detectors in the wavelength ranges of 310-400 nm and 410-650 nm. Therefore, a fluorescence signal can be detected in three channels: FL1 (excitation at 280 nm and emission



at 310-400 nm), FL2 (excitation at 280 nm and emission at 410-650 nm) and FL3 (excitation at 380 and emission at 410-650 nm). The fourth option (excitation at 380 nm and emission at 310-400 nm) is omitted due to elastic scattering. A particle is considered fluorescent if it exceeds the fluorescence threshold in any channel. The threshold $TH$ is obtained by the equation:


$$TH = \overline{FT} + n \cdot \sigma \tag{1}$$

using the forced trigger mode of the WIBS during which the xenon lamps are triggered without any particles present. The mean fluorescence value of each channel $\overline{FT}$ during the forced trigger mode plus a certain factor $n$ of the standard deviation $\sigma$

is calculated for each channel individually and used as the threshold of the channel.

In this data set, a particle is classified as a FAP if it exceeds the threshold calculated with $3\sigma$, whereas it is classified as a highly fluorescent aerosol particle (HFAP) when it exceeds the $9\sigma$ threshold. The three channels can be combined using Perring nomenclature (Perring et al., 2015), where every possible combination of the three channels is considered. Thus, seven types of FAPs and HFPAs, can be defined, as described in Table 1. The maximum concentration is specified as 466 particles

cm⁻³ for FAP counting (10% coincidence) and 9,500 particles cm⁻³ for sizing and counting (10% coincidence). A study on the counting efficiency and size accuracy of the WIBS NEO can be read in Lieberherr et al. (2021). In their study, the specific instrument used, seems to reach an upper detection limit for particle diameter larger than 10 µm. However, the authors comment that this might be specific to this exact instrument. Laboratory experiments with pollen and microplastic particles > 10 µm with the exact WIBS that is used in this study is not consistent with the findings of Lieberherr and colleagues, as

pollen bigger than 40 µm were detected (Gratzl et al., 2024).

Analysis was conducted with the WIBS toolkit (DMT) implemented in IGOR pro 9. Due to the deadtime of the two xenon lamps, a deadtime correction for FAPs is integrated in the toolkit. Therefore, the concentration of excited particles (those which get counted and sized and have their fluorescence intensity measured) is not reported here.

The instrument was attached to an ACTRIS (Aerosol, Clouds and Trace Gases Research Infrastructure) approved, heated total

air inlet. Cloud droplets (and crystals) enter the system but evaporate on the way to the instruments. The inlet was connected to the instrument with conductive tubing (I.D. = 10 mm and L ~ 1.5 m). Although the tubing to the WIBS was kept as vertical as possible, a slight angle of approximately 10° could not be avoided. Consequently, at a low flow rate of 0.3 l/min, we expect great losses for super-coarse (> 10 µm) particles due to sedimentation. Theoretical calculations with the Particle Loss Calculator (Weiden et al., 2009) from Max Planck Institute for Chemistry show particle losses of 7 % at 3 µm and 26 % at 6

µm, respectively, only for the 1.5 m tubing from the end of the TSP inlet to the instrument, assuming a particle density of 1 g cm⁻³ and a shape factor of 1. For the low flow rate of 0.3 l min⁻¹, no values for particle bigger than 6 µm are available by the Particle Loss Calculator. Also connected to the TSP inlet, amongst other instruments, were an aerodynamic particle size (APS, TSI model 3321) and the Portable Ice Nucleation Experiment (PINE, Bilfinger SE, Möhler et al., 2021) The setup is sketched in Fig. 1..



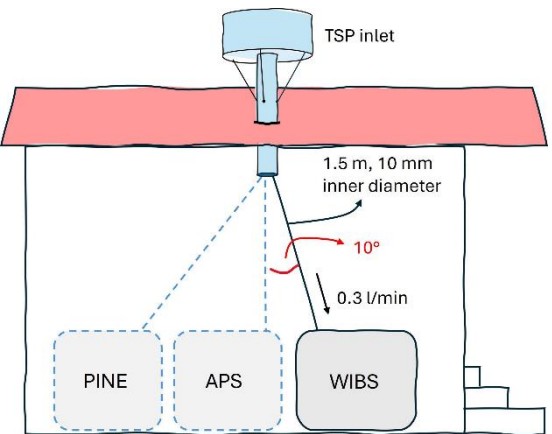


Figure 1. Schematic drawing of the setup in Sammaltunturi station.

Table 1: WIBS particle classification using all possible combinations of the three channels. Biological and nonbiological particles that have been attributed to different channels in laboratory or field studies are listed. References: a: Savage et al., (2017) (laboratory), b: Hernandez et al., (2016) (laboratory), c: Markey et al., (2022), d: Beck et al (2024), e: Gratzl et al., (2024) (laboratory), f: Stone et al, (2021) (laboratory), g: Hughes et al., (2020), h: Mampage et al., (2022), i: Yue et al., (2022), j: Yue et al., (2019), k: Markey et al., (2024), l: Gao et al., (2024), m: Sarangi et al., (2022), n: Fernandez-Rodriguez et al., (2018).

| Particle type | Channels involved | Excitation (nm) | Emission (nm) | Biological contributors | Nonbiological contributors |
|---|---|---|---|---|---|
| A | FL1 only | 280 | 310 – 400 | Bacteria[a,b], Fungal spores[a,b,c], Pollen fragments[a] | Soot[a], Black Carbon[d], Microplastic[e] |
| B | FL2 only | 280 | 420 – 650 | Pollen fragments[a,f,g, h] | Black carbon[a,d], Brown carbon[a], Secondary organic aerosol[i] |
| C | FL3 only | 370 | 420 – 650 | | |
| AB | FL1 and FL2 only | 280 | 310 – 400, 420 – 650 | Fungal spores[a,b,c,j], Pollen fragments[a,e,h] | Microplastic[e] |
| AC | FL1 and FL3 only | 280 370 | 310 – 400, 420 – 650 | | |
| BC | FL2 and FL3 only | 280, 370 | 420 – 650 | Fungal spores[h,j], Pollen[a], Pollen fragments[a,f,g,h], | Black carbon[d,i,k,l] |
| ABC | FL1, FL2 and FL3 | 280 370 | 310 – 400, 420 – 650, 420 – 650 | Fungal spores[b,h,i,j,m,n], Pollen[a,c], Pollen fragments[a,f,g,h] | Microplastic[e] |



## 4 Data evaluation and quality control

The instrument was operated in forced trigger mode every six hours for one minute. The mean values of the forced trigger intensities (and standard deviations) from the four measurements each day (or less, if not all four were measured) was used to calculate the threshold every 24 hours. Over the whole campaign, the threshold calculated from the forced trigger intensity was relatively constant. A time series of the daily $3\sigma$ threshold intensities is depicted in Fig. 2. The standard deviation of the $3\sigma$ threshold mean was highest for the FL1 channel with a relative deviation of $\pm$ 2.8 % from the mean value. The highest

deviation of a single, one day mean value was $\pm$ 9.4 %. The standard deviation for FL2 and FL3 is $\pm$ 1.5 % and $\pm$ 2.3 %, respectively.

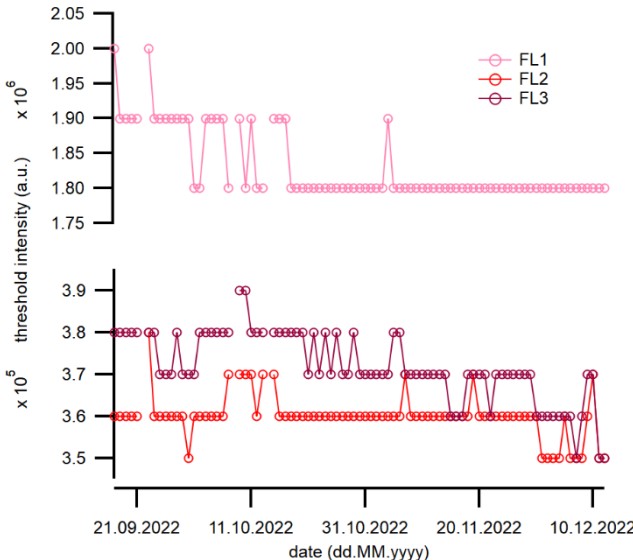

Figure 2: Temporal evolution of the $3\sigma$ threshold intensities for the three channels. A particle was considered fluorescent if its intensity in
any channel was higher than the threshold intensity.

Every particle was detected individually. The dataset presents 30 minutes average values.

This data set contains one file per day with the names TUW-WIBS_N.b1.yyyyMMdd.hhmm.csv, where the date and time indicate the date of the measurement and the time of the first datapoint, respectively. The data set is accessible under the DOI 10.5281/zenodo.13885888 (Gratzl and Grothe, 2024) at the Zenodo Open Science data archive (http://zenodo.org, last access:

18.11.2024). The files include number concentrations of total aerosol particles (TAPs), FAPs and HFAPs and the subtypes of the latter two of diameter 0.50 to 30µm (optical equivalent diameter), as well as number size distributions of TAPs, FAPs and HFAPs, and the subtypes of the latter two of diameter 0.50-30.42 µm (optical equivalent diameter) in 15 logarithmically equidistant size channels: 0.50-0.66 µm, 0.66-0.86 µm, 0.86-1.14 µm, 1.14-1.50 µm, 1.50-1.97 µm, 1.97-2.59 µm, 2.59-3.40 µm, 3.40-4.47 µm, 4.47-5.88 µm, 5.88-7.73 µm, 7.73-10.17 µm, 10.17-13.37 µm, 13.37-17.59 µm, 19.59-23.13 µm, 23.13-

30.42 µm. The concentrations and size distributions are reported in $cm^{-3}$. The variables are *Starttime*: The starting date and





time of the 30 minutes measuring interval (dd.MM.yyyy hh:mm) in UTC, and number concentrations of total aerosol particles: $N\_TAP$, and number concentrations of the following fluorescent particles, all calculated with a 3σ threshold: total FAPs: $N\_FAP\_FL$, FAPs in FL1: $N\_FAP\_FL1$, FAPs in FL2: $N\_FAP\_FL2$, FAPs in FL3: $N\_FAP\_FL3$, FAPs in A: $N\_FAP\_A$, FAPs in B: $N\_FAP\_B$, FAPs in C: $N\_FAP\_C$, FAPs in AB: $N\_FAP\_AB$, FAPs in AC: $N\_FAP\_AC$, FAPs in BC: $N\_FAP\_BC$, and

FAPs in ABC: $N\_FAP\_ABC$, as well as number concentrations of the following highly fluorescent particles, all calculated with a 9σ threshold: total HFAPs: $N\_HFAP\_FL$, HFAPs in FL1: $N\_HFAP\_FL1$, HFAPs in FL2: $N\_HFAP\_FL2$, HFAPs in FL3: $N\_HFAP\_FL3$, HFAPs in A: $N\_HFAP\_A$, HFAPs in B: $N\_HFAP\_B$, HFAPs in C: $N\_HFAP\_C$, HFAPs in AB: $N\_HFAP\_AB$, HFAPs in AC: $N\_HFAP\_AC$, HFAPs in BC: $N\_HFAP\_BC$, and HFAPs in ABC: $N\_HFAP\_ABC$. Further variables are the number size distribution (dN/dlogDp) of TAPs: $SD\_TAP$, the number size distributions of the following

fluorescent particles, calculated with a 3σ threshold. $SD\_FAP\_FL$: total FAPs, $SD\_FAP\_FL1$: FAPs in FL1, $SD\_FAP\_FL2$: FAPs in FL2, $SD\_FAP\_FL3$: FAPs in FL3, $SD\_FAP\_A$: FAPs in A, $SD\_FAP\_B$: FAPs in B, $SD\_FAP\_C$: FAPs in C, $SD\_FAP\_AB$: FAPs in AB, $SD\_FAP\_AC$: FAPs in AC, $SD\_FAP\_BC$: FAPs in BC, $SD\_FAP\_ABC$: FAPs in ABC, as well as the number size distributions of the following highly fluorescent particles, calculated with a 9σ threshold. $SD\_HFAP\_FL$: total HFAPs, $SD\_HFAP\_FL1$: HFAPs in FL1, $SD\_HFAP\_FL2$: HFAPs in FL2, $SD\_HFAP\_FL3$: HFAPs in FL3, $SD\_HFAP\_A$:

HFAPs in A, $SD\_HFAP\_B$: HFAPs in B, $SD\_HFAP\_C$: HFAPs in C, $SD\_HFAP\_AB$: HFAPs in AB, $SD\_HFAP\_AC$: HFAPs in AC, $SD\_HFAP\_BC$: HFAPs in BC, $SD\_HFAP\_ABC$: HFAPs in ABC. Each size distribution variable consists of 15 values per time interval for the 15 size channels. The size channels are indicated as "(lower limit_upper limit)" in µm.

All data was carefully monitored and quality controlled. Missing or outlier datapoints were set to -9999.9. The values of the

first data point of every day were deleted and set to -9999.9, since the data analysis software WIBS toolkit gave too low concentrations, or values of zero, due to the program code. In certain days, the first time stamp of the next day (the deleted one) was also recorded as the last time stamp of the previous day and gave good values. In these cases, these values are used rather than deleting the data from this time stamp. The instrument experienced several breaks throughout the campaign due to software failure. For approximately two hours after restart, the size distributions of certain fluorescence channels were shifted

to smaller sizes with the maximum always being in the lowest size channel. This effect was seen consistently after every break. Therefore, data from the first three hours after a restart was omitted. Although the total concentrations did not seem to be affected, number concentration values were also set to -9999.9 for these periods. Single data points for which the concentration of fluorescent particles was three times higher than the mean value of the three previous and three subsequent datapoints, were examined in detail to rule out artificial spikes. For some datapoints, the higher concentration compared to the neighboring ones

originated from a single, less than two-second-long increase of detected particles by more than a factor of 100, which also showed higher fluorescence intensities. This clearly points to an unphysical event. These datapoints were interpreted as artefacts and the values were set to -9999.9. The origin of these artefacts is not clear.

A size calibration was carried out to test the instruments' ability of accurate particle sizing, before and after the campaign. Before the campaign, Polysterene latex spheres (PSL) of physical diameter of 1 µm were atomized and recorded by the

Earth System
Science
Data

instrument, which detected the peak of the size distribution within 4 % of the physical diameter at 1.04 µm. After the campaign, PSL of 0.8 µm, 1 µm and 2 µm were measured. The instruments response was within 10 % of the physical diameter for 2 µm PSL at 1.8 µm and below 3 % for the remaining sizes (0.995 µm for 1 µm PSL and 0.78 µm for 0.8 µm PSL).

## 5 Dataset Overview

Figure 3 (a) depicts box plots of the concentrations of TAPs, FAPs, and all subtypes of the latter. Fig. 3 (b) depicts box plots of the concentrations of HFAPs and all their subtypes. For all types, the concentrations were divided between 17.09.2022-23.10.2022 and 24.10.2022-13.12.2022 (dd.MM.yyyy) as from the 24.10.2022 the surrounding ground was covered in snow. Snow cover was measured at Kenttärova research station located at 67.987° N and 24.243° E at 347 m above sea level inside a spruce forest approximately 5.5 km east of the Sammaltunturi station. Permanent snow cover leads to a strong decrease in the concentrations of FAPs and especially HFAPs. In Fig. 4, the relative change of the median concentrations from the snow free period to the snow-covered period is plotted for TAP, HFAP_FL and most HFAP subtypes. Whereas the median concentration for TAP is 33 % lower during snow-covered time compared to the snow free period (the mean concentration increases, see Fig. 3) the reduction is more drastic for the highly fluorescent particle types, HFAP_AB, and HFAP_ABC even exceeding 94 % reduction.

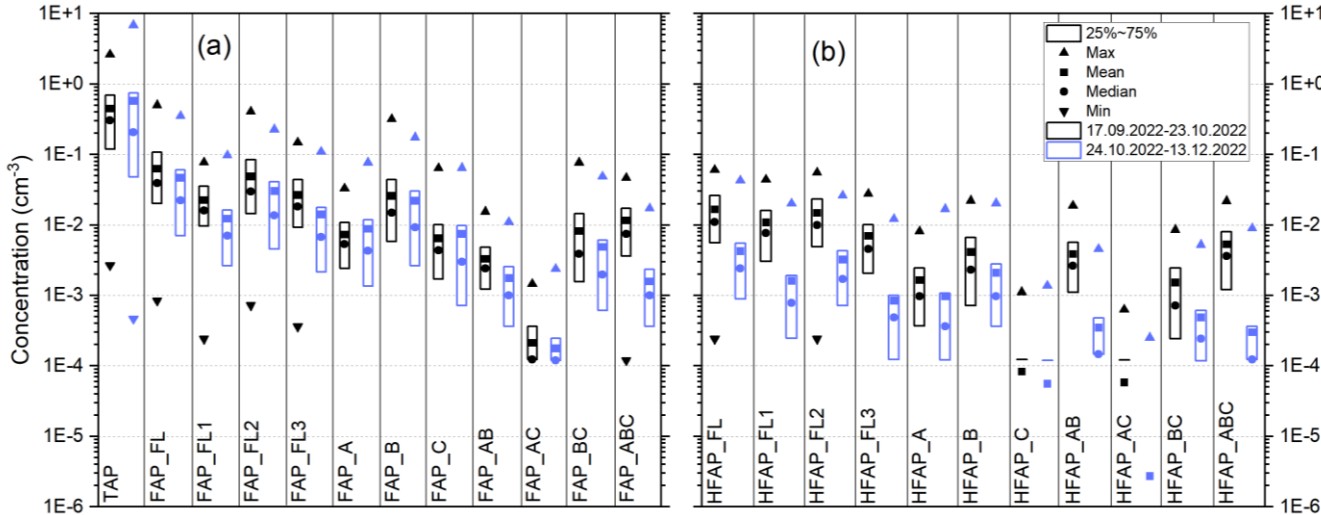

Figure 3. Boxplots of concentrations of (a) TAPs and all subtypes of FAPs and (b) all subtypes of HFAPs. In Black: Concentration from 17.09.2022-23.10.2022 (snow free period of the campaign) and in blue concentrations from 24.10.2022.-13.12.2022 (snow covered period). The horizontal bar in (b) for HFAP_C and HFAP_AC is the 25th percentile.





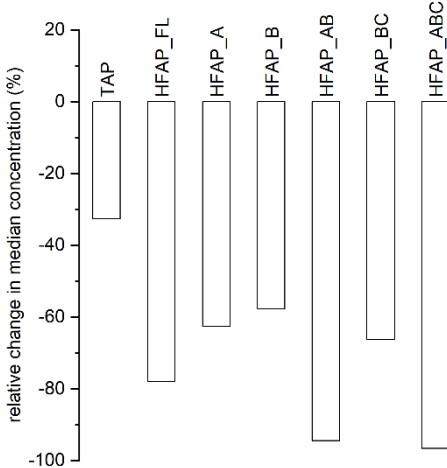

Figure 4. Relative change of median concentrations from the snow free period to the snow-covered period of the campaign for TAP, HFAP_FL and the subtypes of HFAP, except of HFAP_C and HFAP_AC which medians are zero. Mean TAP concentrations decrease by 33 %, while the other particle types decrease more strongly. HFAP_AB and HFAP_ABC decrease the strongest with -94 % and -97 %, respectively.

In Fig.5, the temporal evolution of the number size distributions (dN/dlogDp) for TAPs, and all types of FAPs are plotted in a surface plot. The color code indicates the size distribution dN/dlogDp. For TAP, FAP_A, FAP_B, FAP_C, FAP_AC and FAP_BC the maximum of the distribution is located at the smallest size channel (0.50-0.66 µm). FAP_FL, FAP_FL1, FAP_FL2, FAP_FL3, FAP_AB and FAP_BC all show a bimodal distribution during the snow free period (17.09.2022-23.10.2022). The peak at larger diameters (between 2 and 4 µm) gets less prominent or vanishes completely in the snow-covered period. Only FAP_ABC maintains a single distinct peak for most of the campaign with a maximum at around 4 µm.

A different picture arises when plotting the size distributions of HFAPs and all subtypes of them (see Fig. 6). The higher fluorescence threshold results the loss of most of the FAPs < 1 µm. The bimodal distribution in HFAP_FL and HFAP_FL2 is mainly caused by HFAP_B with a peak maximum at around 1 µm during the snow free period. During the snow-covered period, a peak around 1 µm dominates for HFAP_FL, HFAP_FL1, HFAP_FL2, HFAP_A and HFAP_B. During the snow free period, types HFAP_A, HFAP_AB and HFAP_ABC have a similar size distribution with a maximum at around 4 µm which evolves in a comparable manner. Overall, the decrease in concentration (especially of fine particles < 1 µm) during the snow free period compared to the snow-covered period is much larger for the HFAPs than for the FAPs (Fig. 3, 4 and 5).





Figure 5: Surface plots of size distributions (dN/dlogDp) of TAPs and all types of FAPs. Grey areas refer to values below 0.001 cm⁻³. White areas refer to times where no data was recorded or data that didn't pass the quality check. The left axis (particle diameter) is plotted logarithmically. The underlined date (23.10.2022) indicates the end of the snow free period.




Figure 6: Surface plots of size distributions (dN/dlogDp) of all types of HFAPs. Grey areas refer to values below 0.001 cm⁻³. White areas
refer to times where no data was recorded or data that did not pass the quality check. The left axis (particle diameter) is plotted
logarithmically. The underlined date (23.10.2022) indicates the end of the snow free period.




## 6. Data availability

Datasets are archived under the DOI [10.5281/zenodo.13885888](10.5281/zenodo.13885888) (Gratzl and Grothe, 2024) at the Zenodo Open Science data
archive (http://zenodo.org, last access: 18.11.2024), where a dedicated community Pallas Cloud Experiment – PaCE2022 has been established (https://zenodo.org/communities/pace2022/, last access: 18.11.2024). This community houses the data files along with additional metadata on the datasets from the Pallas Cloud Experiment 2022 campaign.

## 7.Summary

The PaCE2022 campaign was conducted from September to December 2022 in Pallas in the Finnish sub-Arctic. This paper
provides an overview of data on the concentration and size distribution of TAPs, FAPs and HFAPs during this campaign, collected at the Sammaltunturi station, at 565 meter above sea level. The station was located inside clouds about 50 % of the measurement period. In Sect. 2, we shortly describe the research site and the vegetation surrounding the location of the WIBS measurements. In Sect. 3, a short overview of the Instrument is provided. The measured variables and the quality control of the data were explained in Sect. 4.

These are the first such measurements of fluorescent aerosols at the Sammaltunturi research station. They provide valuable insights into the seasonal dynamics of fluorescent and biological particles and how snow cover is influencing their emission. FAPs and HFPAs concentrations can be used as a proxy for PBAPs. The size distributions of the different particle types, together with literature that characterized PBAPs in the laboratory or in field, studied with a WIBS instrument, could help identify different types of PBAPs such as bacteria and fungal spores. Together with meteorological data and air mass
backwards trajectories, the driving forces of the emission of PBAPs from the northern boreal forest can be examined. Caution should be exercised in interpreting particles as pollen grains, as they are usually larger than 10 µm. A size range that is not well covered in this data **set** due to inlet losses. Comparison with other aerosol data, for instance black carbon (see Backman et al., 2025) could give valuable information on interfering particles which are detected as fluorescent but not necessarily originate from the biosphere. As PBAPs might play a crucial role in cloud glaciation processes, this dataset might also be
valuable when compared to INP concentrations, measured with PINE (Böhmländer et al., 2025).

### Author contribution

JG performed the measurements with help from all authors. JG did the data analysis and wrote the manuscript. All authors contributed to the discussion of the data set and reviewed and edited the manuscript. DB and KD prepared and organized the
PACE 2022 campaign. HG and OM funded the journey. HG purchased the WIBS as part of his appointment.

### Competing interests

The authors declare that they have no conflict of interest



**Acknowledgements**

The authors would like to thank the team from Metsähallitus and Sammaltunturi station for their help with logistics and watch over our instrumentation during PaCE 2022 campaign.

**Financial support**

We thank TU Wien and BMBWF for the purchase of the WIBS 5/NEO. We gratefully acknowledge support by the FFG (Austrian Research Promotion Agency) for funding under project no. 888109, the Vienna Science and Technology Fund (WWTF) through project VRG22-003 and TU Wien Bibliothek for financial support through its Open Access Funding Program.

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
