# Peer review of "Fluorescent aerosol particles in the Finnish sub-Arctic during the Pallas Cloud Experiment 2022 campaign"

_Earth System Science Data, 2024_

## Author Comment (AC2)

**Referee 2:**

**General comments:**

This manuscript presents and describes measurements of fluorescent aerosol particles at a remote boreal forest site, accompanied by some interpretative insights. This work has a combination of instrumentation, location and timespan that is rather unique. The manuscript is well written, featuring a comprehensive introduction and an in-depth methodology description. Although the data presentation is concise, it effectively conveys the main observation on the effect of snow coverage on soil emissions of HFAP/FAP. This dataset is particularly valuable to the aeromicrobiology research community, as they contribute to a better understanding of microbiological aerosolization mechanisms. I would recommend the publication of this article but have some minor comments, mostly focusing on how to present the dataset.

We thank Referee 2 for their insightful feedback to our manuscript. Please see our point-bypoint response below, with the points raised by the Referee in black, our responses in blue and the changes made to the manuscript in red.

**Specific comments:**

For lower concentrations of coarse mode aerosols, the recharge time for the Xenon lamps might not impact non-fluorescent but analyzed particle (excited) concentrations significantly. Even then, a comparison between TAP and excited particles could be valuable even as a general percentage. This is specially useful when accessing the disentanglement between coarse mode aerosol and fluorescent aerosol concentrations. Thus, I would suggest presenting this number, alongside concentration numbers and relative numbers for both snow covered and snowfree periods. These metrics could be quite valuable for other researchers using this dataset.

We agree that this could be valuable information. The differences of the excited to total concentration ratio for the snow free and snow-covered period, however, is minor (median values are 98. 51 % and 98.95 %, respectively). Therefore, we think it is sufficient to present this ratio as a median and mean value for the whole campaign. We added the following sentence at line 118:

Due to the generally low aerosol concentrations at the site, the ratio of excited particles to total particles is high, with a median value of 98.76 % (mean: 97.68 %) for the whole campaign.

A table for representing the variables (such as the different types of HFAP and FAP) could be more useful and clearer than a long paragraph.

We agree that this paragraph is hard to read. We made the description of the variables into a table (new Table 2, see below) and deleted the text where we explained each variable.

| Number  | Name       | Description                                       | Threshold |
|---------|------------|---------------------------------------------------|-----------|
| 1       | Starttime  | Starting date and time of the 30 min measuring    |           |
|         |            | interval in dd.MM.yyyy hh:mm (UTC)                | -         |
| 2       | N_TAP      | Conc. of total particles                          | -         |
| 3       | N_FAP_FL   | Conc. of total fluorescent particles              | 3 σ       |
| 4       | N_FAP_FL1  | Conc. of fluorescent particles in FL1             | 3 σ       |
| 5       | N_FAP_FL2  | Conc. of fluorescent particles in FL2             | 3σ        |
| 6       | N_FAP_FL3  | Conc. of fluorescent particles in FL3             | 3σ        |
| 7       | N_FAP_A    | Conc. of fluorescent particles in A               | 3σ        |
| 8       | N_FAP_B    | Conc. of fluorescent particles in B               | 3σ        |
| 9       | N_FAP_C    | Conc. of fluorescent particles in C               | 3σ        |
| 10      | N_FAP_AB   | Conc. of fluorescent particles in AB              | 3σ        |
| 11      | N_FAP_AC   | Conc. of fluorescent particles in AC              | 3σ        |
| 12      | N_FAP_BC   | Conc. of fluorescent particles in BC              | 3σ        |
| 13      | N_FAP_ABC  | Conc. of fluorescent particles in ABC             | 3σ        |
| 14 - 24 | N_HFAP     | Same sequence as variables 3 – 13*                | 9σ        |
| 25      | SD_TAP     | Size distribution of total particles              | 3σ        |
| 26      | SD_FAP_FL  | Size distribution of total fluorescent particles  | 3 σ       |
| 27      | SD_FAP_FL1 | Size distribution of fluorescent particles in FL1 | 3σ        |
| 28      | SD_FAP_FL2 | Size distribution of fluorescent particles in FL2 | 3σ        |
| 29      | SD_FAP_FL3 | Size distribution of fluorescent particles in FL3 | 3 σ       |
| 30      | SD_FAP_A   | Size distribution of fluorescent particles in A   | 3 σ       |
| 31      | SD_FAP_B   | Size distribution of fluorescent particles in B   | 3 σ       |
| 32      | SD_FAP_C   | Size distribution of fluorescent particles in C   | 3 σ       |
| 33      | SD_FAP_AB  | Size distribution of fluorescent particles in AB  | 3 σ       |
| 34      | SD_FAP_AC  | Size distribution of fluorescent particles in AC  | 3 σ       |
| 35      | SD_FAP_BC  | Size distribution of fluorescent particles in BC  | 3 σ       |
| 36      | SD_FAP_ABC | Size distribution of fluorescent particles in ABC | 3 σ       |
| 37 - 47 | SD_HFAP    | Same sequence as variables 26 – 36*               | 9σ        |

Table 2: Description of variables in the data set. Variables 2 - 47 are reported in cm-3. Each size distribution variable (variables 25 - 47) consists of 15 values per time interval for the 15 size channels and is reported as dN/dlogDp. The size channels are indicated as "(lower limit\_upper limit)" in  $\mu$ m in the data set.

\*HFAP refers to particles exceeding the 9  $\sigma$  threshold

Fluorescent aerosols are highly size dependent, smaller particles will seldom present higher than 9 times the background fluorescent signals. Thus, there is not much to gain by presenting fluorescent aerosol timelines by size distribution plots. I would suggest presenting them in total concentration instead. This will make daily cycles (if present) and overall seasonality more easily distinguishable. Or, to have an overlay concentration plot, to not remove the insights deriving from the size distribution (such as the bimodal distribution). Another important aspect that could be explored is presenting the relative contributions of the different FAP/HFAP categories to the overall TAP. This would help disentangle biological emissions from other emissions, given that the mechanisms might not be the same. We thank the referee for this comment. We agree that presenting the concentrations and fractions as timeseries might be more useful than the size distributions. We therefore added a new figure (now Fig. 3, see below) in which we address all the above suggestions:

In Fig. 3 (b), we plotted the concentration of TAPs, FAPs and HFAPs (as 30 min average data points (time resolution of the data set) and a 24 h running median, to guide the eye).

Furthermore, we added the fraction of FAPs and HFAPs in TAPs in Fig. 3 (c).

We feel that presenting all FAP and HFAP types as fractions in TAPs would be overwhelming. Therefore, we present the fraction of particle types in FAPs and HFAPs as stacked fractions. This way, one can easily see the change in contribution of every type (Fig. 3 (d) and (e)).

To better see the change in concentrations from snow free to snow-covered period, we present HFAP\_ABC concentration as an example together with snow depth (Fig. 3 (f)).

Furthermore, to address the next comment, we added wind speed and wind direction data in Fig. 3 (a).

We added the following text:

Figure 3 provides an overview of concentrations of TAP and (H)FAP, fractions of (H)FAP, as well as wind speed, wind direction and snow depth over the whole measurement period. Figure 3 (a) shows wind speed and wind direction data measured at Sammaltunturi station. Figure 3 (b) displays the concentrations of TAP, FAP\_FL and HFAP\_FL. Both FAP\_FL and HFAP\_FL concentrations decrease over time by up to two orders of magnitude, while TAP concentrations exhibit no clear seasonal trend. The fraction of HFAP\_FL in TAP decreases much more substantially over time than the fraction of FAP\_FL (Fig. 3 (c)). Figure 3 (d) and Figure 3 (e), show the stacked fractions of FAP types within FAP\_FL and HFAP types within HFAP\_FL, respectively, revealing changes in the contribution of different particle types. The contribution of AB and ABC particles is much higher at the beginning of the campaign for both cases. ABC (and AB for HFAPs) dominate (H)FAPs until mid of October, after which A and B particles become the dominant contributors for the remainder of the campaign.

From October 24, 2022 the surrounding ground was covered in snow. Snow depth was measured at Kenttärova research station located at 67.987° N, 24.243° E at 347 m above sea level inside a spruce forest approximately 5.5 km east of the Sammaltunturi station. In Fig. 3 (f), the concentration of HFAP\_ABC is plotted alongside snow depth. The data indicates that snow cover has a strong influence on (H)FAP concentrations, particularly for the ABC and AB types: Once the ground is snow-covered, the concentration of HFAP\_ABC decreases rapidly and remains low, with the exception of a brief increase coinciding with a short melting period.

And changed the following paragraph slightly to

Figure 4 shows boxplots of the concentrations of TAPs, FAPs and HFAPs during the snow free period (September 9 – October 23, 2022) and the snow-covered period (October 24 – December 13, 2022). Permanent snow cover leads to a pronounced decrease in the concentrations of FAPs (Fig. 4 (a)) and especially HFAPs (Fig. 4 (b)). In Fig. 5, the relative change of median concentrations from the snow free period to the snow-covered period is plotted for TAP, HFAP\_FL and most HFAP subtypes. While the median concentration for TAP is 33 % lower during snow-covered time compared to the snow free period (the mean concentration increases, see Fig. 3 (a)), the reduction is much more pronounced for the highly fluorescent particle types: HFAP\_AB and HFAP\_ABC both show reductions exceeding 94 %. The influence of other

meteorological variables on HFAP concentrations is discussed in Gratzl et al. (2025), which includes evaluations based on this data set.

We also added the sentence

Wind speed, wind direction and snow depth data were taken from https://en.ilmatieteenlaitos.fi/download-observations (last access: April 11, 2025).

to the data availability statement.

---

## Author Response (AR1)

**Referee 1:**

General comments:

This manuscript by Gratzl et al. summarized the dataset of fluorescent primary bioaerosols using a WIBS, e.g. fluorescence pattern and particle size, based on the intensive observations in Finnish forest site. The dataset showed the significant differences of bioaerosols with seasonal variation, snow-covered or snow-free.

The manuscript is well structured, including descriptions of data quality control, and well written in English. The dataset is generally useful for the researcher and community to work with the biological particles and their impact on the climate. I recommend the publication after the following minor comments are considered.

We thank Referee 1 for their useful comments on our manuscript. Please see our point-by-point response below, with the points raised by the Referee in black, our responses in blue and the changes made to the manuscript in red.

Specific comments:

I understand that there is a difference in fluorescent particles (both normal and highly) between on the snow-free and snow-covered conditions, but is this limited by local emission or not?

I found a description that the site is largely affected by local emissions and surrounded by the biological forest conditions in section 2, but have any additional analysis to evaluate/categorize the local emission or outside contribution, i.e., air mass origin or emission sources? (need more detailed description in line 259 or other part)

We agree that deeper analysis and additional data is necessary to get a comprehensive understanding of the origin of the fluorescent particles. However, we feel that this is outside the scope of this data description paper. In the meantime, we submitted a research article following this paper to Atmospheric Chemistry and Physics that is available as a preprint, in which we address this very question (doi of the preprint: https://doi.org/10.5194/egusphere-2025-1599). We also added a recommendation in the summary, reading

For further analysis it is also recommended to categorize the local emissions or potential far range contributions to FAPs and HFAPs (Gratzl et at., 2025).

Also, do you think how large are contributions from the non-biological but fluorescent particles, as you mentioned in Table 1? Any suggestions?

Previous research has shown that B and BC particle concentrations correlate positively with black carbon concentrations in different environments. We now address this in the summary and added a sentence after the following sentence in the summary:

*"Comparison with other aerosol data, for instance black carbon (see Backman et al., 2025) could give valuable information on interfering particles which are detected as fluorescent but not necessarily originate from the biosphere.":*

For example, B and BC particle concentrations have been shown to correlate with black carbon concentrations in previous field studies (Gratzl et al., 2025; Beck et al., 2024; Gao et al., 2024;

Markey et al., 2024; Yue et al., 2022) and account for approximately 50 % of both FAPs and HFAPs in this data set.

Others:

L36-L49:

It would be useful to add if there is any methods of the detection and identification of bioaerosols as a general introduction, such as offline analysis (e.g. microscopic analysis or biological methods), as well as WIBS and UV-APS (online method).

We would like to kindly point out to the reviewer that we address this already in the introduction right before line 36 in line 28-35:

*"Due to the interaction of PBAPs with human, animal and plant health (e.g. disease transmission, allergic reactions, crop diseases), especially fungal spores and pollen grains have been monitored for decades by aerobiologists using traditional methods like the Hirst trap, first introduced in 1952 (Hirst, 1952). This method relies on the capture of PBAPs on a slowly moving sticky microscope slide and subsequent analysis of pollen grains and fungal spores under an optical microscope. Other commonly used techniques involve examining PBAP concentrations with fluorescence microscopes after DNA staining of PBAPs on filters or by incubation on Agar plates (Després et al., 2012). However, these methods have limited time resolution, and require trained personnel, as well as a high expenditure of time to identify PBAPs. "*

Since none of these offline techniques are part of this data set, we think this paragraph is sufficient to introduce offline techniques for bioaerosol measurements.

L163-176

Why not give one example and briefly summarize the remaining channels as well (likely described in lines 220-222)? I don't think it is necessary to write everything. Or move/associate with the description in the data files as an asset.

We agree that this paragraph is hard to read. Referee 2 suggested creating a table for representing the variables and we think this is also a good solution which will improve overall readability of the paragraph. We deleted the text where we explained each variable, and added the table (new Table 2, see below).

Table 2: Description of variables in the data set. Variables 2 – 47 are reported in cm$^{-3}$. Each size distribution variable (variables 25 – 47) consists of 15 values per time interval for the 15 size channels and is reported as dN/dlogDp. The size channels are indicated as "(lower limit_upper limit)" in μm in the data set.

| Number | Name | Description | Threshold |
|---|---|---|---|
| 1 | Starttime | Starting date and time of the 30 min measuring interval in dd.MM.yyyy hh:mm (UTC) | - |
| 2 | N_TAP | Conc. of total particles | - |
| 3 | N_FAP_FL | Conc. of total fluorescent particles | 3 σ |
| 4 | N_FAP_FL1 | Conc. of fluorescent particles in FL1 | 3 σ |
| 5 | N_FAP_FL2 | Conc. of fluorescent particles in FL2 | 3 σ |
| 6 | N_FAP_FL3 | Conc. of fluorescent particles in FL3 | 3 σ |
| 7 | N_FAP_A | Conc. of fluorescent particles in A | 3 σ |
| 8 | N_FAP_B | Conc. of fluorescent particles in B | 3 σ |
| 9 | N_FAP_C | Conc. of fluorescent particles in C | 3 σ |
| 10 | N_FAP_AB | Conc. of fluorescent particles in AB | 3 σ |
| 11 | N_FAP_AC | Conc. of fluorescent particles in AC | 3 σ |
| 12 | N_FAP_BC | Conc. of fluorescent particles in BC | 3 σ |
| 13 | N_FAP_ABC | Conc. of fluorescent particles in ABC | 3 σ |
| 14 - 24 | N_HFAP_... | Same sequence as variables 3 – 13* | 9 σ |
| 25 | SD_TAP | Size distribution of total particles | 3 σ |
| 26 | SD_FAP_FL | Size distribution of total fluorescent particles | 3 σ |
| 27 | SD_FAP_FL1 | Size distribution of fluorescent particles in FL1 | 3 σ |
| 28 | SD_FAP_FL2 | Size distribution of fluorescent particles in FL2 | 3 σ |
| 29 | SD_FAP_FL3 | Size distribution of fluorescent particles in FL3 | 3 σ |
| 30 | SD_FAP_A | Size distribution of fluorescent particles in A | 3 σ |
| 31 | SD_FAP_B | Size distribution of fluorescent particles in B | 3 σ |
| 32 | SD_FAP_C | Size distribution of fluorescent particles in C | 3 σ |
| 33 | SD_FAP_AB | Size distribution of fluorescent particles in AB | 3 σ |
| 34 | SD_FAP_AC | Size distribution of fluorescent particles in AC | 3 σ |
| 35 | SD_FAP_BC | Size distribution of fluorescent particles in BC | 3 σ |
| 36 | SD_FAP_ABC | Size distribution of fluorescent particles in ABC | 3 σ |
| 37 - 47 | SD_HFAP_... | Same sequence as variables 26 – 36* | 9 σ |

*HFAP refers to particles exceeding the 9 σ threshold

**Referee 2:**

General comments:

This manuscript presents and describes measurements of fluorescent aerosol particles at a remote boreal forest site, accompanied by some interpretative insights. This work has a combination of instrumentation, location and timespan that is rather unique. The manuscript is well written, featuring a comprehensive introduction and an in-depth methodology description. Although the data presentation is concise, it effectively conveys the main observation on the effect of snow coverage on soil emissions of HFAP/FAP. This dataset is particularly valuable to the aeromicrobiology research community, as they contribute to a better understanding of microbiological aerosolization mechanisms. I would recommend the publication of this article but have some minor comments, mostly focusing on how to present the dataset.

We thank Referee 2 for their insightful feedback to our manuscript. Please see our point-by-point response below, with the points raised by the Referee in black, our responses in blue and the changes made to the manuscript in red.

Specific comments:

For lower concentrations of coarse mode aerosols, the recharge time for the Xenon lamps might not impact non-fluorescent but analyzed particle (excited) concentrations significantly. Even then, a comparison between TAP and excited particles could be valuable even as a general percentage. This is specially useful when accessing the disentanglement between coarse mode aerosol and fluorescent aerosol concentrations. Thus, I would suggest presenting this number, alongside concentration numbers and relative numbers for both snow covered and snowfree periods. These metrics could be quite valuable for other researchers using this dataset.

We agree that this could be valuable information. The differences of the excited to total concentration ratio for the snow free and snow-covered period, however, is minor (median values are 98. 51 % and 98.95 %, respectively).  Therefore, we think it is sufficient to present this ratio as a median and mean value for the whole campaign. We added the following sentence at line 118:

Due to the generally low aerosol concentrations at the site, the ratio of excited particles to total particles is high, with a median value of 98.76 % (mean: 97.68 %) for the whole campaign.

A table for representing the variables (such as the different types of HFAP and FAP) could be more useful and clearer than a long paragraph.

We agree that this paragraph is hard to read. We made the description of the variables into a table (new Table 2, see below) and deleted the text where we explained each variable.

Table 2: Description of variables in the data set. Variables 2 – 47 are reported in $cm^{-3}$. Each size distribution variable (variables 25 – 47) consists of 15 values per time interval for the 15 size channels and is reported as dN/dlogDp. The size channels are indicated as "(lower limit_upper limit)" in µm in the data set.

| Number | Name | Description | Threshold |
|---|---|---|---|
| 1 | Starttime | Starting date and time of the 30 min measuring interval in dd.MM.yyyy hh:mm (UTC) | - |
| 2 | N_TAP | Conc. of total particles | - |
| 3 | N_FAP_FL | Conc. of total fluorescent particles | 3 σ |
| 4 | N_FAP_FL1 | Conc. of fluorescent particles in FL1 | 3 σ |
| 5 | N_FAP_FL2 | Conc. of fluorescent particles in FL2 | 3 σ |
| 6 | N_FAP_FL3 | Conc. of fluorescent particles in FL3 | 3 σ |
| 7 | N_FAP_A | Conc. of fluorescent particles in A | 3 σ |
| 8 | N_FAP_B | Conc. of fluorescent particles in B | 3 σ |
| 9 | N_FAP_C | Conc. of fluorescent particles in C | 3 σ |
| 10 | N_FAP_AB | Conc. of fluorescent particles in AB | 3 σ |
| 11 | N_FAP_AC | Conc. of fluorescent particles in AC | 3 σ |
| 12 | N_FAP_BC | Conc. of fluorescent particles in BC | 3 σ |
| 13 | N_FAP_ABC | Conc. of fluorescent particles in ABC | 3 σ |
| 14 - 24 | N_HFAP_… | Same sequence as variables 3 – 13* | 9 σ |
| 25 | SD_TAP | Size distribution of total particles | 3 σ |
| 26 | SD_FAP_FL | Size distribution of total fluorescent particles | 3 σ |
| 27 | SD_FAP_FL1 | Size distribution of fluorescent particles in FL1 | 3 σ |
| 28 | SD_FAP_FL2 | Size distribution of fluorescent particles in FL2 | 3 σ |
| 29 | SD_FAP_FL3 | Size distribution of fluorescent particles in FL3 | 3 σ |
| 30 | SD_FAP_A | Size distribution of fluorescent particles in A | 3 σ |
| 31 | SD_FAP_B | Size distribution of fluorescent particles in B | 3 σ |
| 32 | SD_FAP_C | Size distribution of fluorescent particles in C | 3 σ |
| 33 | SD_FAP_AB | Size distribution of fluorescent particles in AB | 3 σ |
| 34 | SD_FAP_AC | Size distribution of fluorescent particles in AC | 3 σ |
| 35 | SD_FAP_BC | Size distribution of fluorescent particles in BC | 3 σ |
| 36 | SD_FAP_ABC | Size distribution of fluorescent particles in ABC | 3 σ |
| 37 - 47 | SD_HFAP_… | Same sequence as variables 26 – 36* | 9 σ |

*HFAP refers to particles exceeding the 9 σ threshold

Fluorescent aerosols are highly size dependent, smaller particles will seldom present higher than 9 times the background fluorescent signals. Thus, there is not much to gain by presenting fluorescent aerosol timelines by size distribution plots. I would suggest presenting them in total concentration instead. This will make daily cycles (if present) and overall seasonality more easily distinguishable. Or, to have an overlay concentration plot, to not remove the insights deriving from the size distribution (such as the bimodal distribution). Another important aspect that could be explored is presenting the relative contributions of the different FAP/HFAP categories to the overall TAP. This would help disentangle biological emissions from other emissions, given that the mechanisms might not be the same.

We thank the referee for this comment. We agree that presenting the concentrations and fractions as timeseries might be more useful than the size distributions. We therefore added a new figure (now Fig. 3, see below) in which we address all the above suggestions:

In Fig. 3 (b), we plotted the concentration of TAPs, FAPs and HFAPs (as 30 min average data points (time resolution of the data set) and a 24 h running median, to guide the eye).

Furthermore, we added the fraction of FAPs and HFAPs in TAPs in Fig. 3 (c).

We feel that presenting all FAP and HFAP types as fractions in TAPs would be overwhelming. Therefore, we present the fraction of particle types in FAPs and HFAPs as stacked fractions. This way, one can easily see the change in contribution of every type (Fig. 3 (d) and (e)).

To better see the change in concentrations from snow free to snow-covered period, we present HFAP_ABC concentration as an example together with snow depth (Fig. 3 (f)).

Furthermore, to address the next comment, we added wind speed and wind direction data in Fig. 3 (a).

We added the following text:

[revised manuscript text omitted]

It is mentioned in the campaign overview that meteorological (and air mass trajectory) data is available. However it is not presented in the plots. Given that FAP/HFAP are greatly associated with the coarse mode portion of the aerosol population which is mainly wind driven, a wind speed / direction overlay or as another subplot would be greatly appreciated as it would increase the insight on the effect of the snow layer.

We added wind speed and wind direction to (new) Fig. 3, as addressed above. In the meantime, we submitted a research article following this paper to Atmospheric Chemistry and Physics which is available as a preprint, in which we analyze the relations between meteorology and air mass origin with HFAPs in detail (doi of the preprint: https://doi.org/10.5194/egusphere-2025-1599). We added the following sentence at the end of the paragraph in which Figure 4 is discussed (see last comment):

The influence of other meteorological variables on HFAP concentrations is discussed in Gratzl et al. (2025), which includes evaluations based on this data set.

Typeset suggestions:

Line 14 – Intrinsic fluorescence is dependent on size, composition, excitation wavelength and detection limit. I would suggest switching the word "most" to something softer sounding or explicitly mentioning the caveats.

We changed the sentence to "Since biological aerosol particles can exhibit intrinsic fluorescence, ..."

Line 26 – Replace "contains" with "are the".

Done!

Line 59-62 – Sentence starting with "Thus, traditional methods..." seems cluttered. Breaking the sentence down into 2 could improve clarity.

We separated this sentence into two sentences, now reading

Thus, traditional methods of aerobiologists might not be suitable for ice nucleation research, as they only detect intact and big fungal spores and pollen grains with low time resolution. Therefore, these methods potentially overlook important contributors to biological INPs, especially in the size range below about 2 μm in diameter (Fernández-Rodríguez et al., 2018).

Line 67 – Remove "weak" as the main differentiation between biological and non-biological fluorescent aerosol is mainly focused on emission spectra rather than strength.

We removed the "weak".

Line 93 – Is it meant "sheath" flow instead of "sheet"?

Yes, it was supposed to be sheath. We changed it.

Line 151 – Paragraph formatting seems off. Also, it seems like the sentence is disconnected.

We agree and deleted the sentence in line 151.

Line 201 – (dd.MM.yyyy) seems redundant given that values bigger than 12 are present for dd.

We agree! We changed the x-axis label of Figure 2 to "Date" and used the same label for the new Figure 3.

Line 262 – "set" is bold.

It is no longer bold.

**Other Changes:**

We deleted *Black Carbon[d],* from Table 1. as a reference for a nonbiological contributor to A particles, as it was a wrong reference.

We separated the two references at line 403 by a paragraph:

*Hoose, C. and Möhler, O.: Heterogeneous ice nucleation on atmospheric aerosols: a review of results from laboratory experiments, Atmospheric Chemistry and Physics, 12, 9817–9854, https://doi.org/10.5194/acp-12-9817-2012, 2012.*

*Hughes, D. D., Mampage, C. B., Jones, L. M., Liu, Z., and Stone, E. A.: Characterization of atmospheric pollen fragments during springtime thunderstorms, Environmental Science and Technology Letters, 7, 409–414, 2020.*

We increased the font of the labels of Figure 2.

[Figure]

Figure 2: Temporal evolution of the 3σ threshold intensities for the three channels. A particle was considered fluorescent if its intensity in any channel was higher than the threshold intensity.